# A Systematic Review of Executive Function and Information Processing Speed in Major Depression Disorder

**DOI:** 10.3390/brainsci11020147

**Published:** 2021-01-22

**Authors:** Laura Nuño, Juana Gómez-Benito, Viviana R. Carmona, Oscar Pino

**Affiliations:** 1Clinical Institute of Neuroscience (ICN), Hospital Clinic, 08036 Barcelona, Spain; 2Group on Measurement Invariance and Analysis of Change (GEIMAC), Institute of Neurosciences, University of Barcelona, 08035 Barcelona, Spain; juanagomez@ub.edu (J.G.-B.); opino.hbmenni@hospitalarias.es (O.P.); 3Department of Social Psychology and Quantitative Psychology, University of Barcelona, 08035 Barcelona, Spain; viviana.carmona.c@gmail.com; 4Mental Health Center, Hospitalet del Llobregat-Benito Menni CASM, L’Hospitalet del Llobregat, 08902 Barcelona, Spain

**Keywords:** cognitive deficits, executive functions, major depressive disorder, psychomotor speed, neuropsychological profile

## Abstract

Background: Major depression is a psychiatric disorder characterized neuropsychologically by poor performance in tasks of memory, attention, processing speed, and executive function. The aim of this systematic review was to examine the evidence regarding the neuropsychological profile of people with major depression and to determine which of two explanatory models—the processing speed hypothesis or the cognitive effort hypothesis—has most empirical support. Methods: We searched three relevant databases and reviewed the reference lists of the articles retrieved. The results obtained with the Trail Making Test and the Stroop Color-Word Test were reviewed for 37 studies published between 1993 and 2020. Results: The empirical evidence supports both hypotheses: cognitive effort and processing speed, suggesting that depression is not only characterized by psychomotor slowing but also involves a specific deficit in executive function. Discussion: We discuss potentially relevant variables that should be considered in future research in order to improve knowledge about the neurocognitive profile of depression. The main limitation of this study derives from the considerable heterogeneity of participants with MD, which makes it difficult to compare and integrate the data.

## 1. Introduction

Major depression (MD) is a severe psychiatric disorder with an estimated lifetime prevalence in epidemiological studies of between 8% and 16% [1,2,3,4,5]. The WHO ranks depression as the leading cause of disability worldwide, affecting around 322 million people [6].

Cognitive dysfunction is a common feature in people with MD. In fact, people with depression present deficits in various cognitive domains which may explain many of the difficulties that these people experience in daily life, including deficits in attention, executive functioning, psychomotor speed and memory [7,8]. Specifically, some studies have concluded that there is global impairment across a wide range of cognitive domains and that deficits in attention, memory, and executive function are perhaps especially prominent in people with MD [9,10,11]. Other authors suggest a specific impairment of executive functions, affecting the processes of inhibitory control and verbal fluency [7,12]. Impairment in these domains appears to be independent of age and the severity and subtype of depression, and in some cases it persists beyond clinical recovery [7,13]. Finally, authors such as Nilsson et al. suggest that higher cognitive deficits in depression may be secondary to a primary deficit in attention [14]. The basis for this hypothesis is that the proper functioning of executive processes depends on the adequate functioning of more basis processes. It is assumed, therefore, that tasks assessing executive functions implicitly evaluate more basic processes on which the former depend.

Despite numerous advances in knowledge, the mechanism underlying cognitive deficits in MD remains uncertain. There are currently two main hypotheses: the processing speed hypothesis and the cognitive effort hypothesis. The former proposes that a slowing of information processing would have a negative impact on the higher cognitive functions of people with MD [13,15,16]. This hypothesis assumes that cognitive speed is a basic component of higher cognitive functioning, and thus impaired speed would undermine an individual’s performance on tasks that require more complex cognitive processes. Various studies have reported data in support of this hypothesis [15,17,18]. The premise of the cognitive effort hypothesis, by contrast, is that impaired performance among people with MD on tasks requiring higher executive functions is due not merely to generalized slowing of information processing but to a specific executive deficit. This hypothesis distinguishes between automatic tasks and those that require effort. Automatic processes are those that do not require attention and which occur in parallel without interfering with other operations or limiting the capacity of the cognitive system. By contrast, effortful processing requires attentional resources, usually takes place serially, and is influenced by cognitive capacity limitations [19]. A number of studies have found that people with depression show deficits in effortful tasks but not in automatic tasks, thus providing support for the cognitive effort hypothesis [7,19,20,21,22]. The aim of the present systematic review is to analyze the mechanisms underlying the deficits reported by studies that have examined cognitive deficits in people with MD, and more specifically to determine whether these deficits are due to a slowing of psychomotor speed (processing speed hypothesis) or to a specific executive deficit (cognitive effort hypothesis).To this end, we examine the performance of people with MD on the Trail Making Test (TMT) [23] and the Stroop Color-Word test (Stroop Test) [24]. These are commonly administered neuropsychological measures that provides information about visual search speed, scanning, selective attention, processing speed, mental flexibility, and executive functions [17,25,26]. A better understanding of these mechanisms and the variables that influence them would enable a more precise and specific characterization of the cognitive deficits associated with depression, thus providing a platform for the development of more effective psychotherapeutic interventions that reflect the profile of cognitive dysfunction.

## 2. Materials and Methods

### 2.1. Data Collection

Two strategies were used to identify studies for possible inclusion. First, we conducted a systematic literature search in the Medline, Pubmed, and Web of Science electronic databases, using the following key terms: depression OR depressive disorder OR depressive disorders OR major depressive disorder OR depressed OR major depression OR affective disorder (in title), AND Stroop Color Word Test OR Trail Making Test OR Stroop OR TMT (in topic). The search was conducted without language restriction and covered the period between 1960 and 26 April 2020 in Medline and Pubmed, and between 1900 and 28 April 2020 in Web of Science. The same search strategy was applied to all three databases. Second, we also identified potentially relevant studies by examining the reference lists of the articles retrieved through the literature search.

### 2.2. Data Selection

A total of 850 potentially interesting abstracts were identified after removing duplicate studies. Two reviewers working independently inspected the selected abstracts. Next, copies of the full articles were obtained and examined in order to assess their eligibility based on the following inclusion criteria: (a) studies with samples of people with a current or remitted diagnosis of major depression; (b) studies that provided data for the Stroop, TMT, or both; and (c) studies that included healthy control groups.

Articles were excluded according to the following criteria: (a) participants with comorbid disorders, substance use, substance abuse, major depression with psychotic symptoms, bipolar depression, and major depression related to medical illnesses; (b) studies in children, adolescents, or elderly people; (c) studies involving people with high risk of depression or relatives of people with MD; and (d) studies using a modified version of the TMT and/or Stroop test. Meta-analyses and reviews were also excluded because their individual studies and references were inspected and assessed for inclusion in the previous step.

Two independent reviewers carried out the study selection, with any discrepancies being discussed in a consensus meeting involving a third reviewer.

### 2.3. Data Extraction

A coding manual and form were developed. For each study, two reviewers working independently extracted sample and study characteristics for people with MD and controls. Sample characteristics included the proportion of women, mean age, and scores obtained on the Stroop, TMT or both. Illness duration, nature of the sample (inpatients, outpatients), and current stage of diagnosis (acute, remitted stage) were also recorded for people with MD. In cases where illness duration was not reported, we used illness onset as a reference to obtain duration data. Regarding study characteristics, we recorded sample size, publication year, and study location. Inter-rater reliability was as follows: mean Cohen’s [27] k = 0.99 for categorical variables and mean intra-class correlation r = 0.98 for continuous variables. Disagreements were resolved by discussion with a third expert.

### 2.4. Data Synthesis and Analysis

In order to synthesize and interpret the results we took into account the following considerations. Regarding the TMT, if the TMT-A is taken as the baseline for the cognitive skills required by both parts of the test [28], then it is possible to estimate the subject’s performance specifically with respect to the higher-level skills required by the second part (TMT-B). Therefore, we considered that those studies in which the group with MD performed significantly worse than controls on Part A or on both parts of the TMT were studies that provided evidence in support of the processing speed hypothesis. Conversely, studies in which the group with MD performed worse on Part B than Part A, in comparison with controls, were taken as evidence for the cognitive effort hypothesis. Poorer performance on Part B was defined in terms of the difference score, B-A [29]. Regarding the Stroop test, we considered that the first two conditions (Word, Color) reflect cognitive speed for automatic information processing, and thus they involve more basic cognitive skills, whereas the third condition (Color-Word or interference condition) involves suppression or inhibition of the dominant response, and therefore requires more cognitive effort [30]. Studies were regarded as providing support for the processing speed hypothesis if the scores obtained by the group with MD were significantly lower on the first two conditions compared with the third, or on all three conditions, in comparison with controls. Conversely, those studies in which the group with MD performed significantly worse on the interference condition than on the congruent condition, in comparison with controls, were taken as evidence for the cognitive effort hypothesis. Here we also considered the interference score reported in the studies, based on the difference between the predicted and actual score in the incongruent condition.

## 3. Results

A total of 199 articles were retrieved through the database search and examination of reference lists. Of these, 37 met the inclusion criteria for review. Figure 1 provides further details regarding the selection process. The selected articles were published between 1993 and 2020. Fifteen studies included participants experiencing a current episode of MD, and six provided data for a sample of individuals in remission. Thirteen studies reported that participants were outpatients, eight that they were inpatients, and two a mixture of the two. The participants in 14 studies were unmedicated or without treatment at the start of the study, while in 20 studies either part or all of the sample was under pharmacological treatment.

The total sample comprised 1460 people diagnosed with MD and 1357 healthy controls. One study [15] used two control samples: a healthy control group and a group of people with rhinitis. There were also 211 participants with other diagnoses (bipolar disorder, psychotic depression, vascular depression, schizophrenia, and obsessive-compulsive disorder). Among people with depression the mean age was 40.21 years (SD = 9.99), and 56.71% were female. The mean years of education across the 26 studies that provided this information was 12.65 (SD = 2.25), while the mean illness duration across the 17 studies where this was available was 11.40 years (SD = 8.24). Table 1 shows sample characteristics and the main findings for the TMT and the Stroop test.

### 3.1. Studies Reporting Data for the STROOP TEST

Four of the studies analyzed found no differences between people with MD and controls, either when comparing the three conditions and the interference score [54,59,64] or with the number of errors and task completion time [46]. Huang et al. [59] and Crews et al. [54] suggest that these results could be due to sample characteristics such as short illness duration (first episode of MD), a relatively high educational level, which may help to maintain brain function, the type of treatment regime (outpatient or inpatient), and symptom severity.

Several studies report that people with MD performed worse than controls on one or more Stroop conditions. Hasselbalch et al. [58] and Canpolat et al. [53] found a significant difference between people with MD and controls in the incongruent test condition, but in the case of Hasselbach et al. [58] the groups did not differ in their interference score. In four studies [49,52,55,60] the group with MD performed significantly worse than controls on all Stroop conditions. The participants in these studies were mostly outpatients and there was considerable heterogeneity in terms of severity, number of past episodes of MD, and current clinical status.

Degl’Innocenti et al. [17] reported that inpatients with MD performed slower than controls on both Stroop conditions but were not disproportionately slower on the incongruent versus the congruent condition. Participants with MD in the study by Schatzberg et al. [63]) performed significantly worse than controls, but their interference scores were within the expected normal range. In a similar vein, one study [51] with a sample of MD people in remission found a trend towards greater interference in comparison with controls, but this was not statistically significant. Huang [48] and Gohier et al. [56] also found significant differences in response time between people with MD and controls; but, in the case of Gohier et al. [56], the groups did not differ significantly in the number of errors committed. Likewise, the outpatients with MD in the study by Den Hartog et al. [15] performed worse on the Color and Word conditions but not on the Color-Word (incongruent) condition. These findings do not provide evidence of a greater interference deficit in people with MD. Rather, they suggest that slower information processing among people with MD interferes with their performance across all test conditions, consistent with the cognitive speed hypothesis.

In contrast to the above results, seven studies did find a greater interference effect in people with MD compared with controls [44,45,47,50,57,61,62], thus providing support for the cognitive effort hypothesis. It should be noted that five of these studies involved inpatients [50,57,61,62] or mixed inpatient-outpatient samples, aged over 50 [45]. In addition, one study included people with a diagnosis of drug-resistant recurrent MD and long illness duration [44], while another recruited people experiencing a moderate episode of MD with melancholic characteristics [47].

Aside from these findings, some studies also suggest that the cognitive deficits of people with MD are more apparent during the acute phase of the disorder and that they may persist in an attenuated form beyond symptom remission [50,51,58]. It certainly seems that the interference deficit depends on certain clinical and demographic characteristics of people with MD. For example, in the study by Nakano et al. [50], in which two age groups of MD people were compared with controls, age had a significant effect on Stroop performance. Overall, there is evidence of impaired performance among people with MD on the Stroop test, and that they show a deficit in both information processing and in the higher executive functions that process interference. Although the results are inconclusive, the overall analysis suggests that their executive impairment is greater than would be predicted based on their psychomotor impairment.

### 3.2. Studies Reporting Data for the TMT

Seven studies [35,37,42,43,54,59,64] report no significant differences between people with MD and healthy controls on either part of the TMT. It should be noted that these studies included younger participants (in comparison with the other studies in this review), outpatients, those with a first episode of MD or in remission, and unmedicated people. Similarly, one study reporting data for the B-A difference on the TMT found no difference in the B-A score between a group of people with MD and controls [10].

Several studies found that people diagnosed with MD performed significantly worse than controls on both parts of the TMT. Six studies reported that MD groups performed significantly worse on both parts of the TMT [31,38,40,53,55,60]. One study [52] reported a significant difference between the MD group and controls on both parts of the TMT at baseline but not at 3- and 6-month follow up, suggesting that their cognitive function had improved as a result of treatment. Another study of inpatients [33], all with a current major depressive episode, found that the MD group performed significantly worse than controls on both parts of the TMT, suggestive of deficits in psychomotor speed and attentional capacity. Matsubara et al. [36] also reported poorer performance in the MD group on both parts of the test, although there was no significant difference with respect to controls in the B-A difference score. These findings support the processing speed hypothesis.

Two studies also report data for the number of errors committed. Gohier et al. [56]) found that, while people with MD had significantly slower response times compared with controls on both parts of the TMT, the groups did not differ in the number of errors on either part of the test, suggesting that these inpatients had preserved cognitive flexibility. By contrast, Halappa et al. [34] observed a significantly higher number of errors on part B of the TMT among a group of outpatients with MD, suggesting that people with this diagnosis need longer and have to make greater cognitive effort when performing tasks that require inhibitory control.

Other studies suggest that a greater severity of depression is associated with greater cognitive impairment. For example, Preiss et al. [40] found that people with a higher number of previous hospitalizations performed worse on the TMT. In four studies [32,39,57,63] the MD group performed worse than controls on part B of the TMT but not on part A. These studies mainly involved inpatients, notably with melancholic depression and a mean of four or more past episodes. As Harvey et al. [57] suggest, these results indicate the presence of executive dysfunctions, including updating and inhibition processes, and they therefore support the cognitive effort hypothesis.

Four studies reporting data for the B-A difference on the TMT were consistent with the cognitive effort hypothesis. The studies by Hasselbalch et al. [58], Moritz et al. [61], Péron et al. [62] and Salik et al. [41] reported deficits for the MD group on both parts of the TMT, and poorer performance on part B. It should be noted that the studies by Moritz et al. [61] and Péron et al. [62] involved inpatients, including those with a recurrent episode of MD [62], whereas the study by Hasselbalch et al. [58] included remitted people under pharmacological treatment in their first discharge from psychiatric hospital.

Overall, the literature reviewed suggests that people with MD perform worse on the TMT, and that their performance deteriorates with increasing severity of depression [40,52,63]. Although the association between symptom severity and executive performance is not supported by all studies [49], some authors [40,52,54] consider there to be a clear relationship. In summary, although once again there are research findings consistent with both hypotheses, the evidence lends greater support to the cognitive speed hypothesis.

## 4. Discussion

The 37 articles reviewed provide evidence of a clear performance deficit among people with MD on the executive tasks considered. The results obtained appear to lend support both hypotheses, since the samples analyzed generally perform disproportionately worse on test conditions that require greater psychomotor speed and cognitive effort, as compared with controls. Thus, although many studies report poor performance among people with MD on the simpler conditions of the TMT or Stroop test [15,17,33,36,38,58,63], suggestive of psychomotor slowing, this deficit may not be sufficient to explain their poor performance on more complex tasks, which are significantly affected with respect to the simpler test conditions. The studies that report this pattern of results involved in-patients with MD, either in the acute phase or with a recurrent episode. This supports a relationship between clinical severity and test performance.

Overall, the results of the studies do not fully coincide, which may be due to sample heterogeneity with regard to concomitant variables such as the subtype of depression, age, the number of lifetime episodes of depression, and current medication. It is also not known to what extent these variables affect the performance of people with depression on executive tasks, and this makes it difficult to compare results and to opt confidently for one or the other explanatory hypothesis. What this diversity of results does indicate is a characteristic widely observed in people with MD, namely the considerable heterogeneity of their neuropsychological profile. Godard et al. [65] noted this heterogeneity with respect to the nature and extent of cognitive deficits in these people. Regarding executive functions, Stordal et al. [66] defined 56% of people in their group with depression as unimpaired in executive function, although the group as a whole differed significantly from controls on all subtests. These findings raise the question of what variables may account for these differences in the same target population.

### 4.1. State or Trait?

A crucial question that arises from the present review concerns the relationship between depressive symptoms and cognitive deficits in people. Specifically, are the cognitive deficits associated with MD maintained after symptom remission (and thus can be considered a trait of the disorder), or is the cognitive performance of people only affected during acute episodes (i.e., it is state-related)? The studies reviewed suggest that cognitive deficits in people with MD persist—at least in an attenuated form—beyond symptom remission [50,51,58]. Studies using tests other than the TMT or Stroop also support this idea, showing that the improvement in cognitive performance following symptomatic recovery tends to be only partial [13,67,68,69,70,71]. As Hammar et al. [7] note, the residual deficits may represent a vulnerability factor for relapse of MD, although the characteristics of the deficits that persist remain to be elucidated. However, only 54% of the studies reported if the participants were in an acute episode, in remission or were inpatients. More research on this topic is needed to clarify the relation between the evolution of depressive symptoms and cognitive impairment.

### 4.2. Subtypes of Depression

Schatzberg et al. [63] compared a group of people with psychotic depression with a non-psychotic depression group and controls. They found that the psychotic depression group performed significantly worse than the group with non-psychotic depression on both the TMT and the Stroop test. However, it is hard to find additional support for these findings, since studies tend either to use mixed samples of psychotic and non-psychotic depression or to include exclusively the latter. In fact, the majority of studies exclude individuals with psychotic symptoms because of the possibility that these two types of depression may present different cognitive deficits. This aspect could explain part of the variability across studies that examine performance on the TMT and Stroop test, and considering this may help to interpret the data in favor of one or the other hypothesis. In a meta-analysis, Zaninotto et al. [72] concluded that people with melancholic depression showed more severe symptoms and poorer cognitive performance on tasks involving attention, working memory, visual learning, reasoning, and problem solving, in comparison with people with non-melancholic depression.

In summary, the subtype of depression may imply different cognitive deficits, and hence it is a factor that should be more widely considered when conducting research in this field. The specific deficits associated with each subgroup also require more in-depth investigation.

### 4.3. Anxiety

Tarsia et al. [73] suggest that people with mixed anxiety-depression present specific cognitive deficits that are different to those affecting individuals with either depression or anxiety alone. Furthermore, some studies have reported a correlation between anxiety and performance on the TMT or Stroop test, highlighting that anxiety may be a confounding variable when interpreting the results obtained by people with MD. Specifically, Lyche et al. [74] found that people diagnosed with MD and with comorbid anxiety disorder differed significantly from controls on the different conditions of a modified version of Stroop. Similarly, Markela-Lerenc et al. [75] observed that performance on a mixed Stroop task was correlated with the level of anxiety. By contrast, Lemelin and Baruch [76] did not observe this correlation when using a computerized version of the same Stroop test in people with MD. Regarding the data obtained with the TMT, both Smitherman et al. [77] and Misdraji and Gass [78] failed to find a correlation between test performance and anxiety scores. However, their participants were not diagnosed with MD, and hence their results do not provide information about the effect of anxiety on this disorder. It is important to note, therefore, that the studies which report poorer performance among people with MD and anxiety used versions of the Stroop test [74,75], suggesting that this test is more sensitive to the effects of anxiety on performance. However, the extent to which anxiety may affect the performance of people with MD on executive tasks is something that requires further investigation.

### 4.4. Age

Age is a natural variable that affects cognitive performance. Some studies found that age was correlated with TMT performance in both a group with MD and among controls [28,79]. Specifically, one study reported that depressive episodes had an especially negative effect on part A of the TMT, and that this effect was more pronounced among older participants (Mahlberg et al. [79]). Hammar et al. [7], using the Stroop paradigm, and Kertzman et al. [80], using a reverse variant of the Stroop task, also found an association between age and performance on all the test conditions they applied. However, another study found no age-related differences in the Stroop interference score [81].

It is possible that part of the age effect on cognitive deficits in MD is due merely to the course of the disorder. As has been observed with other disorders, the chronicity of depression may itself undermine cognitive functioning, such that older people perform worse on different TMT and Stroop tasks.

### 4.5. Symptom Severity

Various studies have reported less executive impairment in subclinical populations or those without MD [78,82,83,84]. For example, Vanderhasselt and De Raedt [84] found no differences between the performance of remitted people with MD and controls, although EEG findings during task performance did differ between the groups, leading the authors to conclude that deficits in cognitive control increase with each depressive episode and persist after symptom remission. Airaksinen et al. [85] observed impairment associated with MD, but found that minor depression did not affect cognitive performance. In a similar vein, Birinder et al. [86], Cohen et al. [87], Hartlage et al. [19], Preiss et al. [40], and Dong et al. [88] found a correlation between symptom severity and executive performance, thus providing support for the state-related hypothesis. However, Biringer et al. [86] suggested that Stroop interference deficits might not be reversed for up to two years. For their part, Merens et al. [89] found no differences between a remitted group and controls on any of the conditions of a computerized Stroop test.

Several studies have also found no association between task performance and symptom severity [7,49,51,75,76,79,90]. For example, Hammar et al. [7] applied the Stroop paradigm to people in the acute phase of depression and again six months later, when their symptoms had remitted. They found that people continued to differ from controls at follow-up, suggesting that the deficits associated with depression are a trait of the disorder and that they persist despite symptom remission.

The relationship between symptom severity and cognitive deficits in MD requires further elucidation. Furthermore, the use of different scales to measure symptom severity makes it difficult to compare results. A more in-depth review of the methods used to assess depressive symptoms and their relationship to cognitive performance would help to shed light on the question of whether the deficits associated with depression are state-related or a trait of the disorder.

### 4.6. Number of Episodes

Gorwood et al. [91] analyzed data from a sample of 2048 people with depression and found that those with more past depressive episodes had a more severe clinical level of psychomotor retardation and needed longer to perform the TMT. In addition, the differences persisted beyond symptom remission. These findings are consistent with those of Kessing [67], who reported that people with recurrent episodes were significantly more impaired than were those with a single episode. Other studies have likewise found a relationship between the number of past episodes and performance on the TMT or a Stroop task [40,84], although not all reports support this association [49,51,75,92,93]. For example, Talarowska et al. [93] found that people with recurrent depression performed worse than did those with a first episode, but this difference became apparent as early as the second episode; in other words, performance was not associated with the actual number of episodes.

Overall, the findings suggest that the number of past depressive episodes is an important factor to consider in future research, being a sign both of the severity of the disorder and its resistance to treatment.

### 4.7. Medication

The effect of medication on cognitive performance is a complex question that has been widely debated in relation to several severe mental disorders, not only depression [94,95]. One of the main difficulties in interpreting the findings is that studies vary in terms of the dose and type of medication they consider, to which one must add the problem of different degrees of treatment compliance among people groups. The influence of medication on executive performance is thus an aspect that requires further elucidation.

With respect to the relationship between executive performance and treatment response among people with depression, mention should be made of the studies by Pimontel et al. [96] and Sneed et al. [97]. These authors found that people who did not respond to medication scored significantly worse than responders, the conclusion being that treatment response in late-onset depression may be associated with impaired executive function. For their part, Dunkin et al. [98] found a correlation between pre-treatment measures of executive function and subsequent response to fluoxetine, suggesting that subtle prefrontal dysfunction in persons with MD may be predictive of poor response to certain medications. With respect to the effect of different drug treatments on the cognitive performance of people with depression, one study reviewed pharmacological treatments aimed at improving cognition in MD and concluded that monotherapy with certain drugs appears to reduce cognitive impairment [95]. However, the wide variability across studies and some inconsistent results make it difficult to draw clear conclusions, leading the authors to highlight the need for further research into this question.

### 4.8. Limitations

The main limitation of this study derives from the considerable heterogeneity of participants with MD across the studies reviewed. Although our selection criteria sought to achieve as much homogeneity as possible, the results are dependent on the information available in individual studies. Hence, we were unable to consider other variables (e.g., illness duration, number of hospitalizations or past episodes) for selection and analysis because they were not reported in all the studies included in the review. Furthermore, with the aim of ensuring that the associated implications of executive deficits in people with MD were relevant for evidence-based practice, we decided to focus on published studies, rather than also considering gray literature. Finally, since there were not enough data regarding clinical features of the participants, we were not able to establish if there were differences in cognitive performance between individuals with an acute episode of MD and individuals in remission.

## 5. Conclusions

The variable results reported by the large number of studies considered in this review suggest that there may be a bidirectional relationship between MD and neuropsychological deficits, and that both the processing speed hypothesis and the cognitive effort hypothesis may account for certain phases and characteristics of depression. At some stages of the disorder, people may only show higher-level executive impairment, even though there could also be other underlying cognitive deficits. For example, the social and occupational functioning and ability to perform daily living activities of some individuals with mild depressive episodes may appear to be normal. However, they may in fact have cognitive deficits that they manage to compensate for through greater cognitive effort, and consequently their impairment only becomes apparent when asked to perform tasks that imply a high cognitive demand.

More research is clearly needed to consolidate the available findings. At all events, it would be wise, perhaps, to maintain an open mind regarding possible explanatory models, leaving room for an alternative model that considers the neuropsychological profile of depression to be heterogeneous and variable depending on the duration and course of the disorder. Analyzing cross-sectional and longitudinal studies separately and comparing groups of people with different disorder duration would help to determine whether there is a specific pattern of cognitive impairment among people with depression. Thus, for example, cognitive slowing might explain better the impairment on certain automatic tasks during a first episode of depression, whereas at more advanced stages of the disorder the cognitive effort hypothesis would account for the deficits observed on tasks requiring executive processing.

It may be that the wide variety of variables and results reflects a highly variable neuropsychological profile, which can only be captured through the integration of several hypotheses. This highlights the importance of analyzing and controlling for the concomitant variables that may influence the performance of depressed individuals on executive tasks. In this respect, future studies should consider the neuropsychological profile of depression as being dynamic and highly variable depending on the number of episodes, the type of course, and symptom severity, while also paying attention to subtypes of depression and the presence of comorbid disorders. Moreover, it should be considered if the cognitive deficits associated with MD persist after symptom remission (and thus can be considered a trait of the disorder) or if the cognitive impairment in individuals with MD only occurs during acute episodes (i.e., it is state-related). Further research, including controlled trial studies, should be done to continue studying both cognitive performance hypotheses in DM and whether the impairment is a trait of MD or a state related to depressive symptoms.

## Figures and Tables

**Figure 1 brainsci-11-00147-f001:**
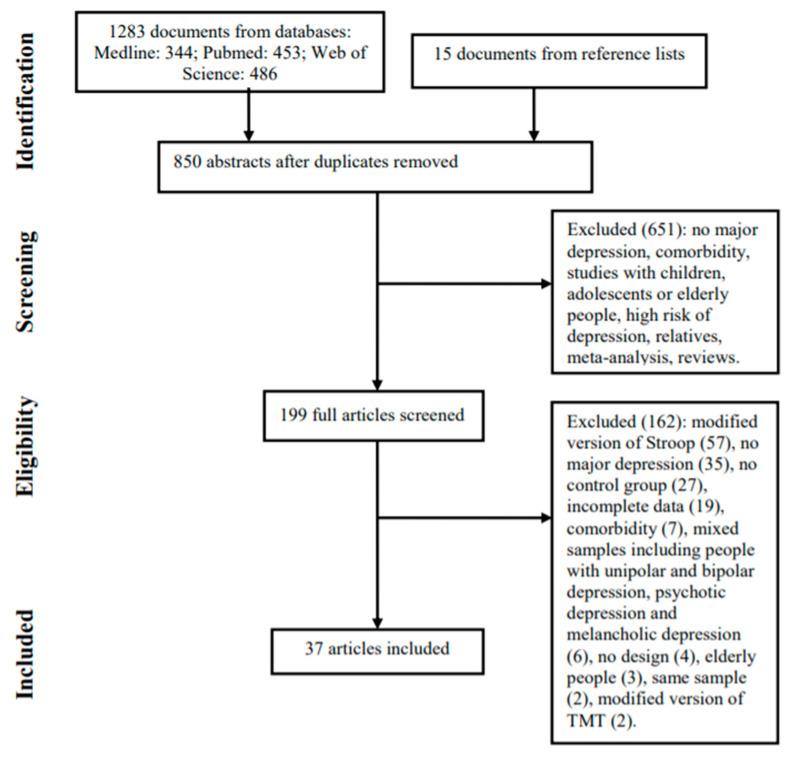
Selection process for study inclusion.

**Table 1 brainsci-11-00147-t001:** Sample characteristics and main outcomes.

	**Participants**		**Trail Making Test**		
**Study**	**Major Depression (MD)**	**Other Diagnoses**	**Sample Characteristics of People Diagnosed with MD**		**Main Outcomes**
Behnken et al. [31]	MD: 20 CG: 20		37.27 mean age, 65% women, 12.13 years of education, 12.77 years of illness duration, 70% of them medicated in past month before the study, in remission		MD < CG on both parts of the TMT.
Cabanel et al. [32]	MD: 34 CG: 29		33.0 mean age, 47.06% women, 11.3 years of education, 76.47% medicated, 100% inpatients		MD = CG on the TMT-A, MD < CG on the TMT-B
Franke et al. [33]	MD: 15 CG: 30	30 people with schizophrenia	42.80 mean age, 100% inpatients in acute episode of major depression.		MD < CG on both parts of the TMT.
Halappa et al. [34]	MD: 65 CG: 19		34.60 mean age, 45.54% women, 11.86 years of education, outpatients.		MD < CG on the TMT-A, MD = CG in the number of errors MD < CG on the TMT-B, MD < CG in the number of errors
Kaczmarczyk et al. [35]	MD: 68 CG: 75		37.4 mean age, 54.41% women, 11.6 years of education, 1.4 years of illness duration, at least 5 days without medication		MD = CG on on both parts of the TMT MD = CG in the B-A difference score.
Landrø et al. [10]	MD: 22 CG: 30		40.60 mean age, 81.82% women, 12.20 of years of education, 100% outpatients and unmedicated, 72.73% with a recurrent episode of major depression.		MD = CG in the B-A difference score.
Matsubara et al. [36]	MD: 17 CG: 27	10 people with bipolar disorder 27 relatives	51.80 mean age, 58.82% women, 13.80 years of education, 8.2 years of illness duration, 100% medicated, 58.82% with a recurrent episode of major depression.		MD < CG on both TMT conditions, MD = CG in the B-A difference score.
Miyata et al. [37]	MD: 65 CG: 67		41.7 mean age, 9.23% women, 16 years of education, 100 medicated, 100% outpatients		MD = CG on both parts of the TMT
Mondal et al. [38]	MD: 30 CG: 30		31.77 mean age, 36.67% women; 100% with at least six years of education, 100% outpatients and unmedicated in a current episode of major depression.		MD < CG on both parts of the TMT.
Moriguchi et al. [39]	MD: 19 CG: 19		40.10 mean age, 42.10 women; 100% unmedicated; 89.47% with melancholic characteristics.		MD = CG in the performance on TMT-A, MD < CG on TMT-B.
Preiss et al. [40]	MD: 97 CG: 97		46.30 mean age, 52.58% women; 16.30 of illness duration in years, 75% medicated, in remission of at least two months.		MD < CG on parts A and B of the TMT. Unmedicated individuals scored higher on the TMT-B. No correlation was observed between performance and the number of depressive episodes, but the number of hospitalizations was correlated with performance on the TMT.
Salik et al. [41]	MD: 60 CG: 30		33.60 mean age, 73.33% women, 100% untreated major depression, outpatients		MD < CG in the B-A difference score
Smith et al. [42]	MD: 48 CG: 48		35.60 mean age, 56.25% women; 100% unmedicated in remission, with at least two previous episodes of major depression		MD = CG on either condition of the TMT. Vortioxetine reduces response time on parts A and B, compared with MD group that received placebo.
Yang et al. [43]	MD: 30 CG: 30	30 people with bipolar disorder	24 mean age, 53.33% women, 9.5 months of illness duration, 100% unmedicated		MD = CG in number of correct, number of errors and response time
	**Participants**		**The Stroop Color-Word Test**		
**Author**	**Major depression (MD)**	**Other diagnoses**	**Sample characteristics of people diagnosed** **with MD**		**Main Outcomes**
Concerto et al. [44]	MD: 11 CG: 11	11 vascular depression	57.18 mean age, 54.55% women, 12.18 years of education, 29.36 years of illness duration, 100% medicated, in a current episode of recurrent and drug-resistant major depression (4 or more episodes).		Significantly greater interference in the MD group compared with CG. MD = CG in the number of errors
Dalby et al. [45]	MD: 22 CG: 22		57.40 mean age, 68.18% women, 13.50 years of education, 100% medicated in a current first episode of major depression, both inpatients and outpatients		MD = CG on the Word condition and the Color condition. MD < CG on the Color-Word condition Significantly greater interference in the MD group compared with CG.
Daniel et al. [46]	MD: 25 CG: 29	25 bipolar disorder	50.60 mean age, 64.00% women, 13.72 years of education, 14.18 years of illness duration, 100% medicated, 100% outpatients		MD = CG with regard to the number of errors and response time
Degl’Innocenti et al. [17]	MD: 17 CG:17		48.20 mean age, 52.94% women, 14.40 years of education, 100% inpatients in the first week of hospitalization.		MD < CG on congruent and incongruent Stroop conditions, but not disproportionately slower on the incongruent condition. Performance was not correlated with symptoms or the severity of the disorder.
Den Hartog et al. [15]	MD: 30 CG1 38 (healthy control) CG2: 20 (severe allergic rhinitis)		41.60 mean age, 46.67% women, 3.3 years of education, 100% unmedicated, 100% outpatients, depressive episode moderate to severe.		MD < CG on automatic conditions, MD = CG on the interference condition (Color-Word).
Gomez et al. [47]	MD: 37 CG: 18		40.89 mean age, 67.57% women, 14.84 years of education, 100% outpatients and unmedicated, all of them with moderate melancholic or endogenous characteristics.		MD < CG on Color-Word condition, significantly greater interference in the MD group compared with CG.
Huang [48]	MD: 19 CG:19		33.16 mean age, 73.68% women, 78.95% above junior school, 100% outpatients and unmedicated in a current episode of major depression.		MD < CG in response time
Lampe et al. [49]	MD: 23 CG: 60		64.00 mean age, 100% women, 11.20 years of education, 31.10 years of illness duration, 100% outpatients with recurrent major depression (euthymic or mild depressive state), 95.65% medicated		MD < CG on all three Stroop conditions No relationship was observed between performance and current level of depression or the number and duration of past depressive episodes.
Nakano et al. [50]	MD: 55 CG: 60	24 participants with MD older than 60 years	45.10 mean age, 41.82% women, 14.20 years of education, 4.60 years of illness duration, 100% medicated, 100% inpatients who were in remission at the time of the study.		Significantly greater interference in the MD group compared with CG.
Paelecke-Habermann et al. [51]	MD: 40 CG: 20		46.28 mean age, 8.74 years of illness duration, 65% medicated, 100% in remission of at least three months. MD people divided into two groups based on their hospital admissions.		MD = CG in the interference score No significant differences were observed with respect to severity.
	**Participants**		**Trail Making Test and Stroop Color-Word Test**		
**Author**	**Major depression (MD)**	**Other diagnoses**	**Sample characteristics of people diagnosed with MD**		**Main outcomes**
Borkowska et al. [52]	MD: 71 CG: 30		44 mean age, 67.61% women, 13.20 years of education, 7 years of illness duration, 100% medicated. Moderate to severe depression	Stroop (congruent &incongruent conditions) Trail Making Test (TMT)	MD < GC on part A and part B of the Stroop test in the baseline assessment and at 3 and 6months after starting treatment. MD < GC on both parts of the TMT at baseline, but there were no differences at 3-and 6-month follow-up.
Canpolat et al. [53]	MD: 41 CG: 44		26.27 mean age, 68.29% women, 12.85 years of education, 100% medicated, in a current episode of major depression.	Stroop (incongruent condition) TMT	MD < CG in the incongruent condition. MD < CG on both parts of the TMT.
Crews et al. [54]	MD: 30 CG:30		20.33 mean age, 100% women, 14.37 years of education, 100% unmedicated and outpatients.	Stroop (three conditions) TMT	MD = CG on the three Stroop conditions. MD = CG on both parts of the TMT.
Dömötör et al. [55]	MD: 71 CG: 99		51.40 mean age, 72.24 women, 3.3 years of education, 46% medicated; 100% outpatients in a current episode of major depression.	Stroop (three conditions) TMT (both conditions)	MD < CG on all three Stroop conditions. MD < CG on both parts of the TMT.
Gohier et al. [56]	MD: 20 CG: 20		41.35 mean age, 75.00% women, 13.12 years of education, 100% medicated, 100% inpatients (within two days of admission, recently started medication).	Stroop (time and errors) TMT	MD < CG in response time, MD = CG in errors committed. MD < CG on both parts of the TMT.
Harvey et al. [57]	MD:22 CG:22		37.4 mean age, 63.64% women, 12.90 years of education, 8.2 years of illness duration, 86.36% medicated, 100% inpatients.	Stroop (three conditions) TMT	MD = CG on the Word condition, MD < CG on the Color and Word-Color conditions. MD = CG on the TMT-A, MD < CG on the TMT-B.
Hasselbalch et al. [58]	MD: 88 CG: 50		59.80 mean age, 68.18% women, 11.90 illness duration in years, 100% medicated, in remission.	Stroop (incongruent condition and interference score) TMT (both conditions and B-A)	MD < CG on the incongruent Stroop condition, MD = CG in the interference score. MD < CG on both parts of the TMT and also in the B-A difference score.
Huang et al. [59]	MD: 25 CG:26		31.40 mean age, 72.00% women, 12 years of education, 12.10 years of illness duration, 100% unmedicated in a first episode of major depression.	Stroop (three conditions and interference score) TMT	MD = CG on the three Stroop conditions and in the interference score. MD = CG on both parts of the TMT.
Krogh et al. [60]	MD: 112 CG: 57		41.60 mean age, 62.50% women, 47.10% high-school education, 100% outpatients, unmedicated, and 87.5% in a current episode of major depression.	Stroop (three conditions) TMT	MD < CG on all three Stroop conditions. MD < CG on both parts of the TMT.
Moritz et al. [61]	MD: 25 CG: 70	25 people with schizophrenia and 25 people with obsessive-compulsive disorder	41.00 mean age, 46.00% women, 11 years of education, 5.6 illness duration in years, 100% inpatients in an acute episode, and 80% of them were medicated.	Stroop (interference score) TMT (both conditions and B-A)	Significantly greater interference in the MD group compared with CG. Cognitive deficits in the MD and schizophrenia groups are comparable. MD < CG on both parts of the TMT and in the B-A difference score.
Péron et al. [62]	MD: 21 CG: 21		49.30 mean age, 71.43% women, 13.30 years of education, 9.8 illness duration in years, 100% inpatients, 100% medicated, 66.67% with recurrent episodes of major depression.	Stroop (interference score) TMT (both conditions and B-A)	Significantly greater interference in the MD group compared with CG. MD < CG on both parts of the TMT and also in the B-A difference score.
Schatzberg et al. [63]	MD: 32 GC: 23	11 people with psychotic major depression	43.10 years, 59.38 women, 15.30 years of education, 100% unmedicated.	Stroop (three conditions and interference score) TMT	Non-psychotic MD = CG on the Word condition. Non-psychotic MD < CG on the Color condition Non-psychotic MD < on the Color-Word condition Non-psychotic MD had poorer interference score (even though their scores fell within the expected normal range). Non-psychotic MD = CG on the TMT-A. Non-psychotic MD < CG on the TMT-B.
Shi et al. [64]	MD: 33 CG: 20		31.52 mean age, 48.48% women, 12.13 years of education, 11.87 years of illness duration, first episode without medication	Stroop (three conditions and interference score) TMT	MD = CG on the three Stroop conditions and in the interference score. MD = CG on both parts of the TMT

Automatic condition of Stroop: mean of Color and Word conditions; CG: Control group; Color-Word: incongruent condition; MD: Major depression; MD < GC: People with MD performed significantly worse than CG; MD = CG: there is no significant difference between MD and CG.

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
