# Peer review of "A Systematic Review of Executive Function and Information Processing Speed in Major Depression Disorder"

_brainsci, 2021, doi:10.3390/brainsci11020147_

Round 1

Reviewer 1 Report

Two hypotheses have been put forward concerning the cognitive effects of major depression (MD): According to some, it reduces the processing speed; others claim that it reduces the capacity of cognitive effort.

The authors’ objective is to decide between these two hypotheses by relying on the data available in the literature on MD. The methodology used for the selection of the 37 relevant studies is rigorous and very clearly stated. In these studies, the processing speed and the cognitive effort of patients with MD are assessed using data from the Tail Making Test and from the Stroop and in comparing them to those of the control group.

The results of these comparisons lead the authors to conclude that MD affects both speed of processing and cognitive effort. They nevertheless analyze the reasons for which the different studies selected are too diverse to draw more precise conclusions. They clearly indicate the limits of their study and identify the different factors that should be controlled to more accurately analyze the cognitive effects of MD.

Author Response

Two hypotheses have been put forward concerning the cognitive effects of major depression (MD): According to some, it reduces the processing speed; others claim that it reduces the capacity of cognitive effort.

The authors’ objective is to decide between these two hypotheses by relying on the data available in the literature on MD. The methodology used for the selection of the 37 relevant studies is rigorous and very clearly stated. In these studies, the processing speed and the cognitive effort of patients with MD are assessed using data from the Tail Making Test and from the Stroop and in comparing them to those of the control group.

The results of these comparisons lead the authors to conclude that MD affects both speed of processing and cognitive effort. They nevertheless analyze the reasons for which the different studies selected are too diverse to draw more precise conclusions. They clearly indicate the limits of their study and identify the different factors that should be controlled to more accurately analyze the cognitive effects of MD.

We really appreciate your feedback, which reinforces our interest in further research in this field

Reviewer 2 Report

The authors reported a systematic review of which the aim was to examine the evidence regarding the neuropsychological profile of people with major depression and to determine which of two explanatory models - the processing speed hypothesis or the cognitive effort hypothesis - has most empirical support. The authors put their attention on studies published between 1993 and 2020 that used the Trail Making Test and the Stroop Color-Word Test.

Even though I have appreciated the effort of the authors in collecting such a big literature (well done!), the aim of the present study is totally unsustainable by the results and from a theoretical point of view.

In my opinion, the crucial question that arises from the present review concerns the relationship between depressive symptoms and cognitive deficits with a focus on cognitive deficits associated with MD maintained after symptom remission vs. cognitive deficits observed only during acute episodes. This latter should be the main aim of the present review; in my opinion, discussing the results about only TMT and Stroop test in relation to the information processing speed (IPS) hypothesis vs. the executive function (EF) hypothesis does not make sense at all. The studies reported were too different and vague to be conclusive: as the authors reported, “overall the results of the studies did not fully coincide, which may be due to sample heterogeneity with regard to concomitant variables such as the subtype of depression, age, the number of lifetime episodes of depression, and current medication. It is also not known to what extent these variables affect the performance of people with depression on executive tasks, and this makes it difficult to compare results and to opt confidently for one or the other explanatory hypothesis”. This theoretical question (the IPS vs. EF hypothesis) should be studied experimental setting or at least carried out within a controlled trial with clinical populations. Moreover, the TMT and The Stroop test both are sensitive to IPS, even though in different ways. For this reason, these tests are not so appropriate to differentiate the two hypotheses (IPS vs. EF). Finally, EF is a really wide theoretical construct that cannot be limited to one or two aspects that are actually reflected in the TMT and The Stroop test. I would suggest to authors to take into consideration the secondary and most intriguing aim for this review (are the cognitive deficits associated with MD maintained after symptom remission (and thus can be considered a trait of the disorder) or is the cognitive performance of people only affected during acute episodes (i.e., it is state-related)?) and change the manuscript accordingly. Maintaining the different MD subtypes should be also a point of interest and strength of this manuscript if modified according to my suggestions. In this present form, this review is not suitable for publication, but changing the aim and taking advantage of all the great work done could be a good way to write a nice piece of evidence in favour of knowledge for the scientific community.

Author Response

Reviewer 2

Even though I have appreciated the effort of the authors in collecting such a big literature (well done!), the aim of the present study is totally unsustainable by the results and from a theoretical point of view.

Thanks for your comment and for appreciating the work done.

In my opinion, the crucial question that arises from the present review concerns the relationship between depressive symptoms and cognitive deficits with a focus on cognitive deficits associated with MD maintained after symptom remission vs. cognitive deficits observed only during acute episodes. This latter should be the main aim of the present review; in my opinion, discussing the results about only TMT and Stroop test in relation to the information processing speed (IPS) hypothesis vs. the executive function (EF) hypothesis does not make sense at all.

We are very grateful for your comment. The focus proposed is very interesting for our research activities. In fact, to analyze if the cognitive impairment occurs only during acute episodes of DM or if it remains after symptom remission was one of our objectives, and we do analyze it in the conclusions section. As it did not have a specific subsection to highlight its relevance, we have enhanced it adding a title to the section where we analyze it (point 4.1. State or trait?, pag 12, on line 294. However, only a proportion of 54% of the studies reported that the participants were in an acute episode, in remission or were inpatients. These data are crucial to explore this issue depper and obtain reliable conclusions. In consequence, we have included it as one of the limitations in our review (line 437).

The studies reported were too different and vague to be conclusive: as the authors reported, “overall the results of the studies did not fully coincide, which may be due to sample heterogeneity with regard to concomitant variables such as the subtype of depression, age, the number of lifetime episodes of depression, and current medication. It is also not known to what extent these variables affect the performance of people with depression on executive tasks, and this makes it difficult to compare results and to opt confidently for one or the other explanatory hypothesis”. This theoretical question (the IPS vs. EF hypothesis) should be studied experimental setting or at least carried out within a controlled trial with clinical populations. Moreover, the TMT and The Stroop test both are sensitive to IPS, even though in different ways. For this reason, these tests are not so appropriate to differentiate the two hypotheses (IPS vs. EF). Finally, EF is a really wide theoretical construct that cannot be limited to one or two aspects that are actually reflected in the TMT and The Stroop test.

Thank you for your comment. Although we used rigorous inclusion criteria in order to achieve a homogeneous sample of studies, we could not reach higher homogeneity because it also depended on the data reported by each of the studies.

 We agree that a controlled trial study would be the most appropriate design to address both hypotheses. Therefore, we have added this suggestion for future research (line 475).

In this review we have used these instruments because they are undoubtedly the most widely used in neuropsychological evaluations. A complete neuropsychological evaluation and careful description of the demographic variables and the characteristics of the disorder would be the most appropriate approach to address cognitive deficits in MD. Unfortunately, reports with these features are scarce and difficult to find in the scientific literature, and the main recognized limitation of any review is that it depends on the studies that have been done in a given field.

I would suggest to authors to take into consideration the secondary and most intriguing aim for this review (are the cognitive deficits associated with MD maintained after symptom remission (and thus can be considered a trait of the disorder) or is the cognitive performance of people only affected during acute episodes (i.e., it is state-related)?) and change the manuscript accordingly. Maintaining the different MD subtypes should be also a point of interest and strength of this manuscript if modified according to my suggestions. In this present form, this review is not suitable for publication, but changing the aim and taking advantage of all the great work done could be a good way to write a nice piece of evidence in favour of knowledge for the scientific community.

Thanks for your suggestion. The question “are the cognitive deficits associated with DM that are maintained after the remission of symptoms or is the cognitive performance of people affected only during acute episode?” it is really crucial. Given the data available to us (in which half of the studies do not report whether the person diagnosed with depression is in the acute phase or in remission), we cannot elaborate on this question extensively, but we have highlighted it as a gap in the field that should be investigated empirically (section 4.1 on line 305), incorporating your suggestions for future research on line 472.

Round 2

Reviewer 2 Report

Even though the authors reply adequately to my comments, I do not fully agree with their response. I can consider, however, the limitation of doing review studies with respect to the data available in each study. This is for sure a huge limitation. Given that the main text cannot be easily changed, and given that the authors provided some core changes to the previous version of the manuscript, I would accept it in its present form. Nonetheless, I strongly suggest to the authors to change the title to be more informative about the content of the paper. One example would be as follow: A Systematic Review of executive function and information processing speed in Major Depression disorder.

Author Response

We are grateful for your reply and suggestion. We really appreciate the careful review of our article which has helped us improve it and thinking about future challengues. We also agree in changing the title (line 1).

Thank you
